



**Technical note: Identification of chemical composition and source of fluorescent**
**components in atmospheric water-soluble brown carbon by excitation-emission**
**matrix with parallel factor analysis: Potential limitation and application**
Tao Cao[1,2,3], Meiju Li[1,2,3], Cuncun Xu[1,2,3], Jianzhong Song[1,2,5,*], Xinjun Fan[4], Jun Li[1,2,5],
Wanglu Jia[1,2], Ping'an Peng[1,2,3,5]
[1]State Key Laboratory of Organic Geochemistry and Guangdong Provincial Key Laboratory
of Environmental Protection and Resources Utilization, Guangzhou Institute of
Geochemistry, Chinese Academy of Sciences, Guangzhou 510640, China
[2]CAS Center for Excellence in Deep Earth Science, Guangzhou 510640, China
[3]University of Chinese Academy of Sciences, Beijing 100049, China
[4]College of Resource and Environment, Anhui Science and Technology University,
Fengyang 233100, China
[5]Guangdong-Hong Kong-Macao Joint Laboratory for Environmental Pollution and Control,
Guangzhou 510640, China
*Correspondence to:* Jianzhong Song, E-mail: songjzh@gig.ac.cn.



**Abstract**

Three-dimensional excitation-emission matrix (EEM) fluorescence spectroscopy is an important method for identification of occurrences, chemical composition, and sources of atmospheric chromophores. However, current knowledge on identification and interpretation of fluorescent components is mainly based on aquatic dissolved organic matter and might not be applicable to atmospheric samples. Therefore, this study comprehensively investigated EEM data of different types of strong light-absorbing organic compounds, water-soluble organic matter (WSOM) in different aerosol samples (combustion source samples and ambient aerosols), soil dust, and purified fulvic and humic acids by an EEM-parallel factor method. The results demonstrated that organic compounds with high aromaticity and strong electron-donating groups generally present strong fluorescence spectra at longer emission wavelength, whereas organic compounds substituted with electron-withdrawing groups have relatively weaker fluorescence intensity. In particular, aromatic compounds containing nitro groups (i.e., nitrophenols), which show strong absorption and are the major component of atmospheric brown carbon, exhibited no significant fluorescence. Although fluorescent component 1 (235, 270/330 nm) in ambient WSOM is generally considered as protein-like groups, our findings suggested that it is mainly composed of aromatic acids, phenolic compounds, and their derivatives, with only traces of amino acids. Principal component analysis and Pearson correlation coefficients between mass absorption efficiency at 365 nm ($MAE_{365}$) and humification index (HIX), C1, C2, and C3



indicated that the highly aromatic and oxidized fluorescent component 3 may be an
important contributor to the light-absorption capacity of ambient WSOM. These
findings provide new insights for the analysis of chemical properties and sources of
atmospheric fluorophores using the EEM method.

**1. Introduction**

Brown carbon (BrC) is a class of organic compounds with light absorption in

near-ultraviolet and visible wavelength, and is ubiquitous in ambient aerosols, cloud
or fog, and rainwater (Fu et al., 2015; Laskin et al., 2015). Owing to its strong
light-absorption capacity, BrC can cause up to 45 % solar radiation absorption by
atmospheric aerosols, and have potential effects on regional and even global climate
(Zhang et al., 2013). In addition, BrC also participates in atmospheric photochemical
reactions, affects the physicochemical properties of atmospheric aerosols (Laskin et
al., 2015; Tang et al., 2020b), and potentially, can be activated to form reactive
oxygen species that cause adverse effects on human health (Bates et al., 2019; Cao et
al., 2021).

Excitation-emission matrix (EEM) fluorescence spectroscopy is a highly

sensitive and widely used analytical technique for the identification of chemical
characteristics and sources of chromophores in dissolved organic matter (DOM) in
aquatic environments (Murphy et al., 2010, 2013; Zhang et al., 2014). Recently,
EEM has been further extended and frequently applied for the investigation of
water-soluble organic matter (WSOM), such as light-absorbing organic compounds





in atmospheric aerosols and fine particles from combustion process (Chen et al.,
2020; Fan et al., 2016; Wu et al., 2020). For instance, humic-like substances (HULIS)
and protein-like substances (PRLIS) have been identified as important fluorescent
components in combustion-derived particles and ambient aerosols (Cao et al., 2021;
Matos et al., 2015; Wu et al., 2020). Chen et al. (2016) used EEM coupled with
parallel factor analysis (PARAFAC) and high-resolution mass spectrometry to
identify chromophores in ambient aerosols, and proposed that fluorescent
components with longer excitation (Ex)/emission (Em) wavelengths comprise more
highly oxygenated groups (Chen et al., 2016b). In addition, further application of the
EEM method has also revealed that the concentration and types of fluorophores
obviously vary during atmospheric processes, such as photolytic aging of biomass
burning(BB)-derived chromophores (Aftab et al., 2018; Fan et al., 2020; Tang et al.,
2020b). Therefore, the EEM method has significant potential for the characterization
(types, sources, and evolution) of atmospheric BrC.
However, application of the EEM method for the identification of atmospheric
BrC has some limitations. It is well known that the present identification,
classification, and interpretation of fluorescent components in atmospheric WSOM
are mainly based on the fluorescence peak position of DOM in aquatic environments
(Coble, 1996; Wünsch et al., 2019). Nonetheless, the chemical and molecular
composition and source of WSOM in atmospheric aerosols significantly vary from
those of WSOM in aquatic environments (Graber and Rudich, 2006; Laskin et al.,
2015); hence, the current fluorescence criterion derived from aquatic environments



could lead to some inaccurate description of the fluorescent components in
atmospheric WSOM. For instance, the EEM region at Ex/Em = 235(270)/330 nm is
assigned to PRLIS and/or tryptophan-like substances in aquatic environment (Coble,
1996), but is also associated with non-nitrogen species such as polyphenols in
atmospheric WSOM (Chen et al., 2016b). The EEM region at peak M (290–
315/370–420 nm) is considered as a typical signal of marine-derived HULIS (Coble,
2007; Zhao et al., 2019), but the source of this peak should be cautiously
investigated when interpreting BrC in continental aerosols. In addition, the
intensities of fluorescent species are not always linearly correlated with their
concentrations, which can be affected by the aromatic ring system, number and types
of functional groups, and inner-filter effects (IFEs), thereby leading to a greater
uncertainty in intensity measurements (Andrade-Eiroa et al., 2013; Chen et al., 2020;
Wang et al., 2020). Atmospheric BrC is composed of complex organic molecules
with various properties (Chen et al., 2020), and only a subset of the BrC molecules
that contain functional groups are capable of fluorescence emission upon relaxation
from an excited state (Andrade-Eiroa et al., 2013). Hence, interpretation of
fluorescence data may only correspond to fluorescent chromophores and may not be
representative of BrC as a whole (Chen et al., 2020; Wang et al., 2020). All these
factors limit further application of the EEM method for the analysis of atmospheric
BrC. Therefore, it is essential to investigate the light-absorbing species that can be
detected by EEM and obtain important information for identifying the chemical
compositions and possible sources of these species.



Accordingly, in the present study, the EEM profiles of a series of BrC model
compounds and WSOM isolated from primary combustion samples, soils, and
atmospheric aerosols were investigated. The chemical characteristics and sources of
the main fluorophores were interpreted according to the fluorescence location and
intensity, and the chemical structure of the model compounds and source samples
analyzed. Then, atmospheric aerosols in Guangzhou (GZ) and Chuzhou (CZ) cities
were collected and fluorescent chromophores within the water-soluble fraction were
identified to estimate the application of the EEM-PARAFAC method in
characterizing atmospheric BrC. The results obtained help to broaden application of
the EEM-PARAFAC method to study atmospheric BrC.

**2. Materials and methods**
2.1. Materials
For accurate identification of the chemical composition and structures of
fluorophores in atmospheric BrC and for assessment of application of the EEM
method to examine atmospheric BrC, a total of 136 samples were investigated in this
study. The samples comprised: (1) 35 BrC model compounds, including phenolic
compounds, aromatic acids, nitroaromatic compounds (NACs), PRLIS,
N-heterocyclic compounds, and polycyclic aromatic hydrocarbons (PAHs) and their
derivatives (Table S1). These compounds are usually detected in ambient samples
and have been considered as the typical BrC model compounds; (2) 13 primary
combustion source samples collected from biomass burning, coal combustion(CC),



and vehicle emission (VE); (3) five soil samples obtained from the rural area of
Guangdong Province, China, with different vegetation; (4) six purified fulvic and
humic acids (FAs and HAs, respectively) kindly provided by Professor Weilin Huang
(Rutgers, The State University of New Jersey, NJ, USA); and (5) 34 diurnal fine
particulate matter ($PM_{2.5}$) samples collected from 6 to 22 April, 2021 at GZ and CZ,
respectively. In addition, 43 annual $PM_{2.5}$ samples were collected from February
2018 to January 2019 at the GZ site and classified as wet and dry season
atmospheric $PM_{2.5}$ samples (for detailed information, see Test S1 of supporting
information (SI)).

**2.2. Standard solution and aqueous extraction of ambient samples**

Solutions of model organic compounds were prepared by dissolving a certain

amount of dried solid or liquid samples in Milli-Q water or methanol. The ambient
aerosol and soil samples were ultrasonically extracted with ultrapure water for three
times, and the supernatants were filtered using a 0.22-μm PTFE syringe filter to
isolate the WSOM. The specific separation and purification methods have been
published in previous studies (Chen et al., 2020; Fu et al., 2015; Wang et al., 2020;
Yan and Kim, 2017) and presented in SI (Test S2).

**2.3. EEM-PARAFAC analysis**

The EEM fluorescence spectra of the aqueous extraction of the samples in 1-cm

quartz    cuvettes    were    recorded    using    a    three-dimensional    fluorescence
spectrophotometer (Aqualog; HORIBA Scientific, USA) at room temperature. The



scanning ranges for Ex and Em was 200–500 and 250–550 nm, respectively. The
wavelength increment of the Ex and Em scans was 5 nm, integration time 0.5 s, and
Milli-Q water (18.2 $\Omega$) used as blank reference. The absorbance measurements were
used to correct the EEM for IFEs as described previously (Fu et al., 2015) if the
absorbance was > 0.05 at 250 nm (Murphy et al., 2013; Tang et al., 2020a).
Background samples were also analyzed and the background values were subtracted
from the values obtained for all the samples. To avoid concentration effects, the
fluorescence spectra were normalized by the water Raman area to produce Raman
Unit (R.U.) and further by organic carbon concentration of the samples to the
normalized fluorescence intensities (R.U./(mg C/L)) (Yang et al., 2022) are shown in
Table S2.

The PARAFAC modeling procedure was conducted in MATLAB 2014b

(Mathwork.Inc, USA) using the drEEM toolkit (Murphy et al., 2018; Wünsch et al.,
2019). The PARAFAC was computed using two to nine component models, with
non-negativity constraints and residual analysis, and split-half analysis was
employed to validate the number of fluorescent components. Based on the results of
the split-half and core consistency analyses, three-component models were chosen
for further investigation. The relative contribution of individual chromophores was
estimated by calculating the maximum fluorescence intensities ($F_{max}$: maximum
fluorescence intensity of the identified fluorescent components; relative content (%)
= $F_{max}/\Sigma F_{max}$) (Chen et al., 2020; Fan et al., 2020).




## 3. Results and discussion

### 3.1. Fluorescence properties of BrC model compounds

To identify whether the light-absorbing species possess fluorescence, a series of BrC model compounds were tested by the EEM method, and the fluorescence profiles are shown in Fig. S1. The results revealed that the location and intensity of the fluorescence peaks of different compounds were different, which varied with the distinct functional groups and aromatic conjugate system.

Although phenolic compounds are important light-absorbing species in atmospheric BrC (Smith et al., 2016; Yu et al., 2014, 2016), not all of them exhibit strong fluorescence. As shown in Fig. S1a, a strong fluorescence peak in the EEM spectrum of phenol was observed at Ex/Em = 270/295 nm. When the phenol compounds were substituted with electron-donating groups (e.g., hydroxyl), all of the stronger fluorescence peaks were obviously red-shifted to 310–320 nm (e.g., catechol, hydroquinone, and 2-methoxyphenol). However, phenolic compounds substituted with electron-withdrawing groups (e.g., carboxyl and aldehyde) displayed weaker or even no fluorescence (Fig. S1a). These differences could be owing to the ability of the electron-donating groups to form a larger conjugate system coupled with the benzene ring and decrease the π→π* transition energy, thus leading to an increase in the Em wavelength (i.e., red shift) and variation in fluorescence intensity. In contrast, the electron-withdrawing group can reduce the conjugate structure formed by the benzene ring and hydroxyl group, reducing the



fluorescence intensity (Andrade-Eiroa et al., 2010; Andrade-Eiroa et al., 2013).

Aromatic acid and its derivatives are also important light-absorbing organic

compounds in atmospheric BrC. Owing to the negative effects of carboxyl group, a
weak fluorescence peak (275/315 nm) was identified for benzoic acid and no
fluorescence was detected for benzene polycarboxylic acids, such as phthalic acid,
terephthalic acid, and trimesic acid (Fig. S1b). However, when benzoic acid was
substituted with electron-donating groups (e.g., hydroxyl, methoxy), higher intensity
fluorescence peaks were observed. Two strong fluorescence peaks at 230/405 and
290/405 nm were identified for 2-hydroxybenzoic acid substituted with only one
hydroxyl group. These peaks could have been the result of the ortho structure of the
hydroxy and carboxyl groups, which is favorable for the formation of intramolecular
hydrogen bond and generates a double-ring conjugate structure, reducing the
transition energy and thereby presenting strong UV absorption and fluorescence
(Andrade-Eiroa et al., 2013).

The N-containing compounds, especially NACs, have strong light absorption,

and have been reported to be the major components of atmospheric BrC, accounting
for more than 60 % of the total light absorption intensity at 300–500 nm (Huang et
al., 2021; Lin et al., 2016; Lin et al., 2017). However, most of the NACs did not
exhibit any fluorescence (Fig. S1c), similar to that reported in a previous study by
Chen et al. (2020), which could be owing to the significant reduction in the electron
density of benzene ring by the nitro ($-NO_2$) group—strong electron-withdrawing
group—thereby weakening the fluorescence.



Tryptophan and tyrosine are the two most studied PRLIS species, and their

EEM spectra are generally used as standards for comparison with fluorophores in

atmospheric WSOM and aquatic DOM (Coble, 1996, 2007). As shown in Fig. S1d,

the Ex/Em peaks at 275/300 and 275/350 nm corresponded to tyrosine and

tryptophan, respectively. The maximum Em wavelength of phenylalanine was more

inclined to short wavelength (280 nm) and with much weaker fluorescence intensity.

Moreover, the fluorescence peaks of PRLIS were obviously overlapped with phenols

and aromatic acids (Fig. S1a, b). It must be noted that the concentrations of phenols

and aromatic acids were significantly higher than those of tryptophan and tyrosine in

the atmospheric samples (Table S2); therefore, the aerosol BrC fluorophores in these

regions are more likely to have originated from phenols and aromatic acids rather

than PRLIS.

The N-heterocyclic compounds such as pyrrole, pyridine, and imidazole are

commonly identified in atmospheric samples (Dou et al., 2015; Jiang et al., 2019;

Kosyakov et al., 2020). However, no fluorescence was observed for these three

species in the present study, indicating that the absorbed energy may have been

consumed by relaxation or vibration (Fig. S1e). Nevertheless,

imidazole-2-formaldehyde produced two strong fluorescence peaks at 290/440 and

350/440 nm, formed from the oxidation of imidazole, suggesting that some

N-heterocyclic compounds from secondary reaction may exhibit strong fluorescence

at higher wavelength in atmospheric BrC (Ackendorf et al., 2017).

PAHs and its derivatives are mainly formed from incomplete combustion



processes, and are important components of BrC (Chen et al., 2020; Lin et al., 2017;
Mahamuni et al., 2020). As shown in Fig. S1f, all PAHs exhibited strong
fluorescence emission, with its peak location associated with the conjugated
aromatic system. Naphthalene presented a fluorescence band located at the
maximum Em wavelength of about 325 nm. As expected, with the increasing size of
the π-bond system and degree of conjugation, the fluorescence band moved toward
the longer wavelength range, and a new Em band was observed at 360–390 nm for
3–4-ring phenanthrene and pyrene, and at 400–500 nm for ≥ 5-ring PAHs
(Mahamuni et al., 2020). The fluorescence spectra of high-ring PAHs were more
complex because of more types of double bonds. As shown in Fig. S1f, the intensity
and location of the fluorescence peaks were also significantly changed when
different types of groups were substituted with PAHs. For example, 1-naphthol
exhibited stronger EEM peak at a relatively longer wavelength (230, 290/460 nm)
owing to its high conjugated structure, when compared with naphthalene. This EEM
spectrum was located in the EEM region of FAs, implying that FAs are composed of
aromatic units and O-containing groups. In contrast, relatively weaker fluorescence
was observed for 9-fluorenone, anthraquinone, and 2-naphthalenecarboxylic acid,
and no EEM signals were observed for 2-nitronaphthol (Fig. S1c), which was
substituted with a strong electron-withdrawing group ($-NO_2$).

**3.2. Fluorescence properties of BrC from different sources**

As shown in Fig. S4a and S4b, BB and CC WSOM exhibited similar



fluorescence spectra, with two types of fluorescence peaks at Ex/Em ≈ (230–
240)/(340–400) nm (peak A) and Ex/Em ≈ (260–280)/(330–360) nm (peak B),
respectively. The two fluorescence peaks were similar to those previously reported
for BB WSOM and HULIS (Fan et al., 2020; Tang et al., 2020a; Yang et al., 2022).
In general, peak A mainly corresponds to the protein-like UV region, with a minor
contribution from fulvic-like substances, whereas peak B could be attributed to
tryptophan-like fluorophores. However, based on the results of the present study,
these two peaks could be mainly attributed to aromatic species such as aromatic
acids, phenolic compounds, and minor quantity of PAHs (e.g., naphthalene) (Fig. 1).
The fluorescence spectra of WSOM from two types of vehicles (diesel and gasoline)
also presented two fluorophores. A relatively strong fluorescence peak was observed
at the low Ex wavelength (Ex/Em ≈ 230/350 nm) and a relatively weaker peak was
detected at the high Ex wavelength (Ex/Em ≈ 270/350 nm) (Fig. S4c). These results
are consistent with those reported in previous studies on VE (Chen et al., 2020; Tang
et al., 2020a; Yang et al., 2022), and similar to the EEM fluorescence spectra of BB
and CC WSOM. However, the fluorescence ranges of vehicles WSOM were
obviously narrower, suggesting that BB and CC WSOM fluorescent components are
more complex.

Soil-derived DOM is also a primary source of atmospheric WSOM. As shown

in Fig. S5a, two main fluorescence peaks located at Ex/Em = 230/430 and 320/430
nm, respectively, were detected in the fluorescence spectra of soil DOM, which are
similar to those reported in previous studies (Ge et al., 2021; Liu et al., 2009) and



particularly close to the position of FAs (Fig. S5b).

Secondary chemical formation is another important source of atmospheric

WSOM. For example, the aqueous-phase reactions of aldehydes with ammonium
sulfate (AS) can produce highly fluorescent species (Hawkins et al., 2016).
Glyoxal-AS and glyoxal/glycine reaction products fluoresce at 340/450 nm, whereas
formaldehyde-AS reaction product fluoresce at 250/430 nm. Secondary organic
aerosols (SOAs) produced in the limonene/$O_3$ system have been reported to strongly
fluoresce in the presence of $NH_3$ (Bones et al., 2010). In addition, aging of primary
organic compounds has also been found to change the fluorescence spectra (Lee et
al., 2013; Li et al., 2021; Powelson et al., 2014). For instance, aging of syringic acid
with OH radicals caused the initial fluorescence band to move toward the long
wavelength range, producing a new band at a broad Em band at 400–600 nm.
Similarly, the fluorescence peaks red-shifted (e.g., from 260–270/360 nm to 280–
290/390–400 nm) during the $O_3$ aging process (Fan et al., 2020), suggesting the
degradation of the initial compound and formation of new secondary organic
compounds generally located at longer wavelengths, possibly with a high degree of
aromaticity or highly oxidized functional groups (Chen et al., 2016a).

**3.3. Identification of chemical species and potential sources of fluorescent**
**components in ambient aerosols**

The typical EEM spectra of atmospheric water-soluble light-absorbing

compounds are shown in Fig. 2. Three fluorescence peaks were identified in the





aerosol WSOM samples: a stronger fluorescence peak at Ex/Em = 230–250/360–420
nm, and two relative weaker fluorescence peaks at Ex/Em = 270–290/340–370 nm
and 300–320/360–420 nm, respectively. Similar fluorescence bands have been
previously identified in the EEM fluorescence spectra of WSOM from $PM_{2.5}$ in
Xi'an of Northwest China and in Godavari of Nepal (Qin et al., 2018; Wu et al.,
2019). Although the fluorescence intensities varied with different sites and seasons,
the EEM spectra of WSOM were very similar, making it difficult to directly
distinguish the different samples solely based on the characteristics of the EEM
profiles. Therefore, a more powerful protocol named the PARAFAC method was
employed to identify the individual fluorophores in ambient WSOM.

**3.3.1. Identification and quantification of fluorescent components by the**
**PARAFAC method**
To better explain the various fluorophores in different atmospheric WSOM
samples, the EEM spectra were resolved with the EEM–PARAFAC tool
(Andrade-Eiroa et al., 2013; Murphy et al., 2013). As shown in Fig. 3, three
fluorescent components (C1, C2, and C3) were identified in the atmospheric samples;
C1 occurred at relatively lower Em wavelength, exhibiting two fluorescence peaks at
Ex/Em = 235(270)/330 nm, C2 presented fluorescence peaks at around Ex/Em =
235(320)/390 nm, and C3 had a longer Em wavelength than C1 and C2, which was
located at Ex/Em = about 250(355)/455 nm. In general, these fluorescent
components have been interpreted based on the knowledge of fluorescence


characteristics of aquatic DOM. Accordingly, C1 is considered to belong to the
typical PRLIS (Chen et al., 2016a; Chen et al., 2020), C2 is associated with
fulvic-like substances or less-oxygenated HULIS (Chen et al., 2016a; Wang et al.,
2020), and C3 is usually considered to correspond to terrestrial HULIS that are
highly oxygenated organic matter (Table S3) (Chen et al., 2016a; Zhou et al., 2017).
However, it must be noted that the sources and transformation process are
significantly different for WSOM in aerosols and DOM in aquatic environment;
therefore, the fluorescence classifications of DOM might not be applicable to
atmospheric WSOM.
In general, the Ex and Em wavelengths of fluorescent components are mainly
associated with their chemical characteristics and structures (Table S1 and Fig. 1). In
the present study, C1 was similar to tryptophan-like fluorophore associated with
PRLIS in rainwater (Zhang et al., 2014; Zhou et al., 2017) and fog water (Bianco et
al., 2014; Bianco et al., 2016). However, this fluorophore might also be possibly
related to the small molecular aromatic compounds, such as aromatic acids (e.g.,
3,5-dihydroxybenzoic acid and 2-naphthalenecarboxylic acid) and PAHs (e.g.,
naphthalene, phenanthrene, and anthraquinone) (Fig. 1a) (Miyakawa et al., 2015; Wu
et al., 2019). In addition, this fluorophore could also contain traces of some phenolic
compounds, including catechol, hydroquinone, and 2-methoxyphenol. These organic
species might have probably been generated by various types of combustion process
and atmospheric oxidation reaction. It must be noted that investigations of the
fluorescent components in atmospheric WSOM should not only consider their



position in the fluorescence spectrum, but also their concentration and possibility of
trapping. Many previous studies have reported that the concentration of amino acids
in the atmospheric aerosols is almost negligible, when compared with that of lower
molecular weight aromatic compounds such as aromatic acids and phenolic
compounds (Table S2). Therefore, fluorescent components in this Ex/Em region
could be attributed to non-nitrogen aromatic species, rather than PRLIS. Moreover,
this fluorophore overlapped with that of WSOM from combustion process such as
BB, CC, and VE (Fig. 1b), suggesting significant contribution of combustion
process.
When compared with C1, C2 exhibited a strong fluorescence peak at longer
Ex/Em wavelength of 235(320)/390 nm, implying that this fluorescent component
presented relative larger molecular size and higher aromaticity than C1 (Pöhlker et
al., 2012). As shown in Fig. 1a, the fluorescence of C2 is similar to that of aromatic
compounds  (e.g.,  2-naphthalenecarboxylic  acid,  2-hydroxybenzoic  acid,
anthraquinone) and high-ring PAHs (e.g., pyrene, anthraquinone, anthracene,
chrysene) (Mahamuni et al., 2020), and overlaps with the fluorescence spectra of
FAs. In addition, this fluorophore has also been reported to be related to the
generation of SOAs from organic precursors emitted from biological/anthropogenic
emission  and  combustion  process  (Wang  et  al.,  2020).  For  example,  the
aqueous-phase reactions of aldehydes with AS has been proposed as an important
source of atmospheric BrC, which present similar fluorescence spectral profiles
(Hawkins et al., 2016; Lee et al., 2013) (Fig. 1b). Besides, oxidative oligomerization



of phenols and their derivatives can also shift the Ex/Em wavelength of these
substances to longer wavelength, falling into similar fluorescence region (Li et al.,
2021; Tang et al., 2020a; Vione et al., 2019). As suggested by Chen et al. (2016a),
this fluorescent component may be a less-oxygenated fluorescent group contributed
by biomass combustion. Therefore, fluorophore C2 might be related to the
derivatives of biomass burning and/or biogenic molecules, with relatively lower
degree of oxidation (Chen et al., 2016a; Jiang et al., 2022).
C3 presented longer Em wavelength than C1 and C2, with two peaks at around
Ex/Em=250/455 nm and 355/455 nm (Fig. 3). This fluorescent component overlaps
with the fluorescence of high-ring PAHs and their derivatives, such as fluoranthene,
benzo-b-fluoranthene, benzo-a-pyrene, indeno-123cd-pyrene, 1-naphthol, and
N-heterocyclic compounds, including imidazole-2-formaldehyde (Chen et al., 2020;
Mahamuni et al., 2020). Furthermore, this fluorescent component exhibited similar
Ex/Em wavelength to that of FAs and HAs (Fig. S5b), suggesting the possible
contribution of soil dust, and thus could be assigned as HULIS (Lin and Guo, 2020).
Similar fluorescent substances have also been identified in the study of atmospheric
aerosol fluorescent chromophores, such as "HULIS-1" at about 470 nm in Nagoya,
Japan (Chen et al., 2016a) and "component 2" at about 480 nm in Godavari, Nepal
(Wu et al., 2019). Based on the PARAFAC results with aerosol mass spectrometry
data, C3 was considered to be a fluorescent group with high oxygen content and high
O/C ratio, close to that of aged organic aerosols (Chen et al., 2016a; Jiang et al.,
2022) (Fig. 1b). It must be noted that low molecular weight organic compounds can





further undergo oligomerization to high molecular weight species with long Em
wavelength during the aging process (Hawkins et al., 2016; Li et al., 2021; Tang et
al., 2020b; Yu et al., 2016). The resulting compounds may present a more complex
structure than its precursor, probably owing to the presence of condensed aromatic
ring and other π-electron systems with a high level of conjugation; thus, atmospheric
aging is assumed to be a potential contributor to C3 (Barsotti et al., 2016; De
Laurentiis et al., 2013; Hawkins et al., 2016).

**3.3.2. Spatial and seasonal variations of fluorescent components in WSOM**

The relative contributions of C1, C2, and C3 components to the total

fluorescence intensities ($F_{max}/\sum F_{max}$) were calculated (Fig. 4), and were found to be
similar for WSOM from CZ and GZ, exhibiting maximum C2 content and relatively
lower C1 and C3 contents. Furthermore, these WSOM samples showed obvious
spatial and seasonal variations. First, CZ WSOM presented relatively higher C3
content, whereas GZ WSOM had relatively higher C2 content at the same sampling
period. Such differences in the composition of fluorescent components may be
ascribed to the variation in the primary emission sources and atmospheric aging
process in the two sites. The relatively higher C3 content in CZ could be attributed to
the comparatively high contribution of soil dust in the suburban region. In contrast,
the relatively higher C2 content in GZ WSOM may be attributed to the
comparatively stronger atmospheric chemical reaction associated with bio-volatile
organic compounds (bio-VOCs) in the hot and humid region of GZ. This result was



416 consistent with the relatively higher humification index (HIX) and normalized

417 fluorescence volume (NFV) values (log(NFV)) of CZ WSOM (Fig. S6) (Chen et al.,

418 2020; Yang et al., 2022).

419  In addition, the resolved Ex and Em spectra for GZ WSOM were also similar in

420 different seasons, implying that the types of fluorophores contributing to WSOM

421 were predominantly the same throughout the year. However, the compositions of

422 fluorescent components varied in different seasons. In the dry season (October–

423 March), WSOM showed relatively higher contents of C3 fluorophores, whereas in

424 the wet season (April–September), slightly higher contents of C2 fluorophores were

425 detected (Fig. 4) (Chen et al., 2020; Wang et al., 2020). These differences might

426 possibly be associated with the variations in the source composition and aging

427 effects of BrC in different seasons. The higher content of C3 in WSOM in the dry

428 season suggested the occurrence of more highly aromatic and highly oxidized

429 compounds. These results could be explained by the fact that more aged organic

430 aerosols and dust were transported from the northern region of China (Jiang et al.,

431 2021). In contrast, the slightly higher C2 content in the wet season may be attributed

432 to the relatively stronger secondary formation of bio-SOAs and photodegradation

433 effects in high-temperature and relative humidity season.


435 **3.4. Correlation with optical and fluorescence properties**

436  It is well known that some organic substances that can generate fluorescence

437 should absorb energy and transit from ground state to excited state and back to





ground state, which could produce corresponding fluorescence. However, it is very
difficult to clearly understand the absorption spectra and corresponding fluorescence
spectrum of complex WSOM. As indicated in the present study, not all WSOM with
strong light absorption exhibit strong fluorescence (e.g., NACs). Hence, to elucidate
the association between light-absorbing chromophores and fluorescent components,
the relationship between optical properties and fluorophores was analyzed by using
principal component analysis and Pearson correlation analysis (Fig. S7). The
MAE$_{365}$ showed an obvious positive loading for principal component 1 (PC1) and
was grouped with C3 and HIX. Moreover, the MAE$_{365}$ values were positively
correlated with C3 and HIX, indicating that light absorption by BrC is more
dependent on these fluorophores with long Ex and Em wavelengths and high
humification. These results also suggested that the oligomerized fluorescent products
with high aromaticity and oxygenated groups may be the dominant factor affecting
the light absorption capacity of the fluorescent chromophores. Similar findings have
also been reported by Chen et al. (2020) and Tang et al. (2021).

**4. Conclusion and future prospects**
In this study, the fluorescence properties of BrC model compounds were
investigated to determine the chromophoric species that can be evaluated by the
EEM method. Accordingly, the aerosol WSOM in two sites (CZ and GZ) were
investigated by the EEM-PARAFAC method, and the chemical characteristics and
potential sources of fluorescent components were examined. The main conclusions



and future prospects are as follows:

(1) Fluorescent components have predominantly been evaluated based on the

knowledge of fluorophores in aquatic DOM, which often leads to misinterpretation.
In the present study, the chemical characteristics of fluorophores in different Ex/Em
regions were discussed based on the fluorescence properties of BrC model
compounds and their amounts in aerosols. In particular, the C1 fluorophore in
atmospheric WSOM, which has been frequently assigned to PRLIS because of the
similarity in fluorescence spectra, was demonstrated to mainly include aromatic
acids, phenolic compounds, and their derivatives, with negligible amount of amino
acids.

(2) The fluorescence properties of target compounds are mainly influenced by

the aromatic system and characteristics of adjacent functional groups. Organic
compounds with high aromaticity and strong electron-donating groups (e.g.,
hydroxyl, methoxyl) generally exhibited strong fluorescence spectra at longer Em
wavelength, whereas organic compounds substituted with electron-withdrawing
groups presented relatively weaker fluorescence intensity. In particular, aromatic
compounds containing nitro groups (i.e., nitrophenols) showed strong absorption and
were the major component of atmospheric BrC; however, they did not exhibit
significant fluorescence. Thus, fluorescence method could only measure a subset of
chromophores in aerosol BrC and should be used with caution for the investigation
of aerosol BrC.

(3) The EEM spectra for aerosol WSOM were very similar; however, the



relative contents of certain fluorescent components significantly varied with the
sampling site and season. For example, more fluorescent components associated
with dust and secondary oxidation of small molecular compounds from combustion
emission were identified in GZ WSOM, whereas more fluorescent components
derived from atmospheric chemical reaction of bio-VOCs were observed in CZ
WSOM. In addition, GZ WSOM exhibited more highly aromatic and highly
oxidized compounds in the dry season.
Although many studies have applied EEM-PARAFAC method to investigate
atmospheric WSOM and have obtained useful data, there are still challenges and
gaps that must be addressed. First, caution should be taken for credible
interpretations of the fluorescent components in atmospheric WSOM because of the
differences in chemical characteristics of organic matter derived from different
sources. In addition, the same fluorophores may exhibit different Ex/Em ranges and
intensities in different environmental conditions (e.g., pH, co-existing metal ions and
inorganic salts, etc.). Therefore, more theoretical and experimental studies are
necessary to understand the relationship between the fluorescent groups and
positions of fluorescence peaks, as well as the influences of sources and chemical
formation process of the fluorescent groups on fluorescence peaks.

**Data availability**
The research data can be accessed in the Harvard Dataverse (https://
doi.org/10.7910/DVN/ULCIU9, Song, 2022)

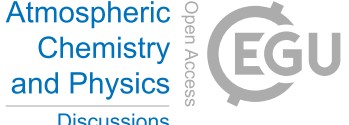


**Author contributions.** J. Song designed the research. T. Cao and C. Xu, analyzed
the model compounds and WSOM samples by UV-Vis and EEM. T. Cao and X. Fan
resolved the EEM by PARAFAC tool. M. Li carried out the $PM_{2.5}$ sampling
experiments. T. Cao and J. Song wrote the paper. J. Li, W. Jia, and P. Peng
commented and revised the paper.

**Competing interests.** The authors declare that they have no conflict of interest

**Acknowledgments.** The present work was supported by the National Natural
Science Foundation of China (42192514 and 41977188), Guangdong Foundation for
Program of Science and Technology Research (2020B1212060053), and Guangdong
Foundation for Program of Science and Technology Research (2019B121205006).

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




**(a)**

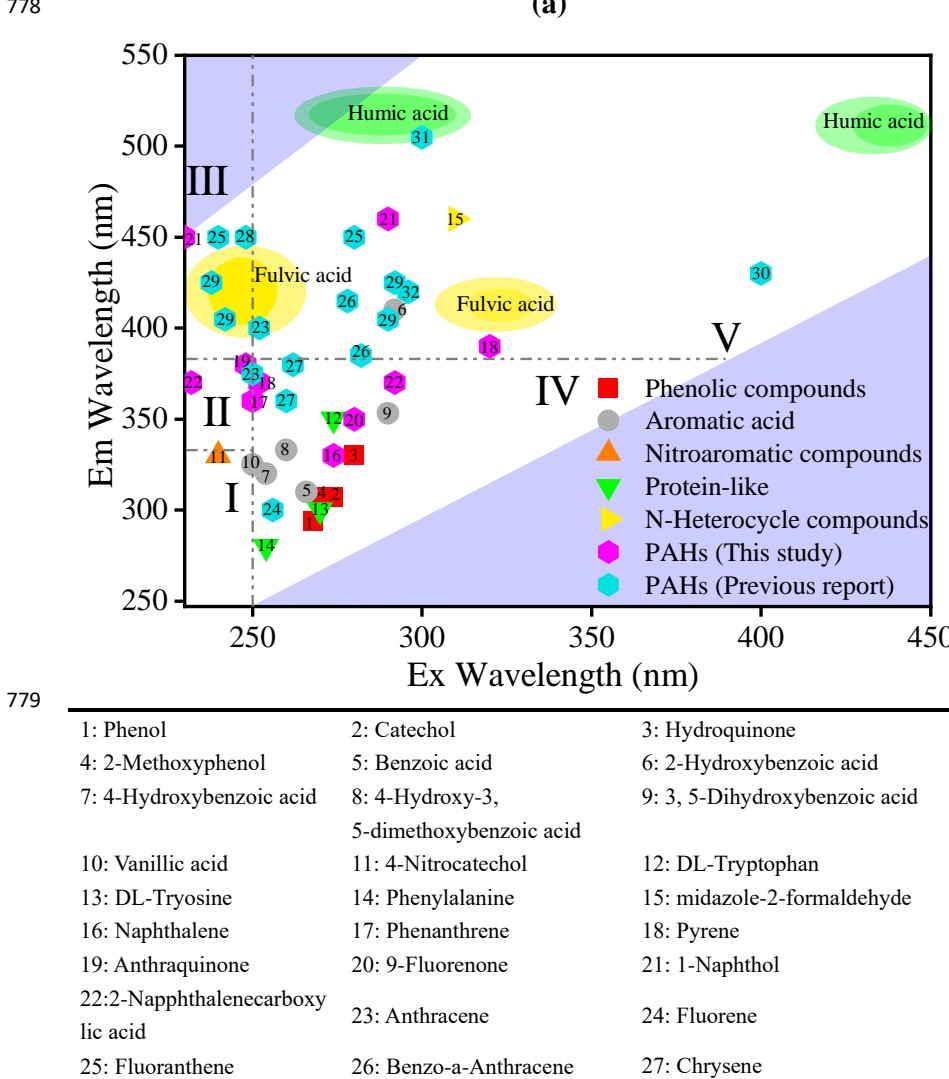


| | | |
|---|---|---|
| 1: Phenol | 2: Catechol | 3: Hydroquinone |
| 4: 2-Methoxyphenol | 5: Benzoic acid | 6: 2-Hydroxybenzoic acid |
| 7: 4-Hydroxybenzoic acid | 8: 4-Hydroxy-3, 5-dimethoxybenzoic acid | 9: 3, 5-Dihydroxybenzoic acid |
| 10: Vanillic acid | 11: 4-Nitrocatechol | 12: DL-Tryptophan |
| 13: DL-Tryosine | 14: Phenylalanine | 15: midazole-2-formaldehyde |
| 16: Naphthalene | 17: Phenanthrene | 18: Pyrene |
| 19: Anthraquinone | 20: 9-Fluorenone | 21: 1-Naphthol |
| 22:2-Napphthalenecarboxy lic acid | 23: Anthracene | 24: Fluorene |
| 25: Fluoranthene | 26: Benzo-a-Anthracene | 27: Chrysene |
| 28: Benzo-b-Fluoranthene | 29: Benzo-k-Fluoranthene | 30: Perylene |
| 31: Indeno-123cd-Pyrene | 32: Benzo-ghi-Perylene | |





**(b)**

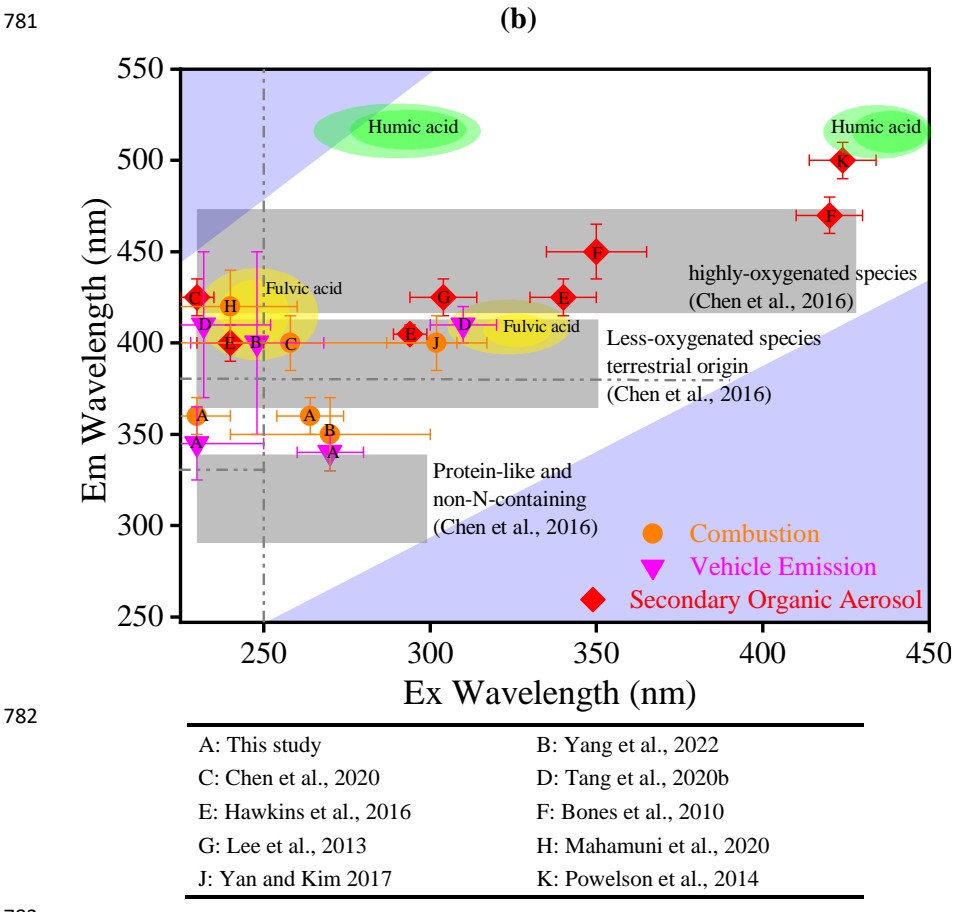


| A: This study | B: Yang et al., 2022 |
|---|---|
| C: Chen et al., 2020 | D: Tang et al., 2020b |
| E: Hawkins et al., 2016 | F: Bones et al., 2010 |
| G: Lee et al., 2013 | H: Mahamuni et al., 2020 |
| J: Yan and Kim 2017 | K: Powelson et al., 2014 |


**Figure 1.** Comparison of chemical characteristics of molecules assigned to each
fluorescence component of BrC model compounds (a) and source WSOM (b).







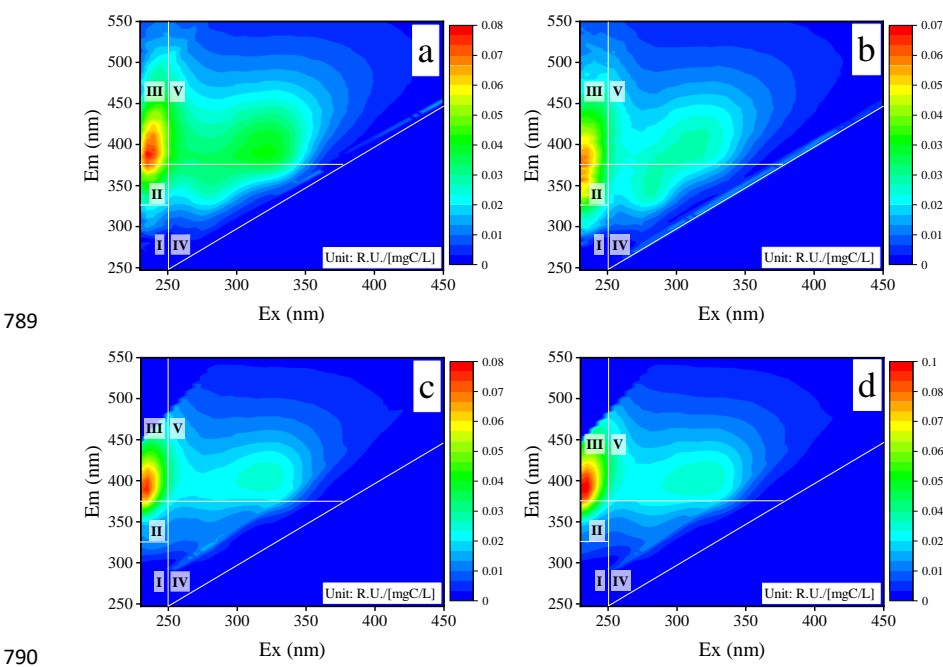



**Figure 2.** The 3D-EEM spectra of WSOM in atmospheric PM2.5 samples (a: Chuzhou (CZ); b: Guangzhou (GZ); c: GZ wet season; d: GZ dry season)








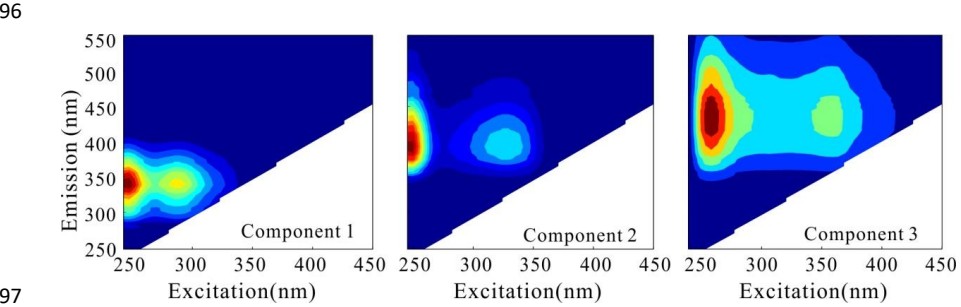



**Figure 3.** The EEM components derived from the PARAFAC model of WSOC in atmospheric PM$_{2.5}$ samples collected at Chuzhou (CZ) and Guangzhou (GZ) sites.







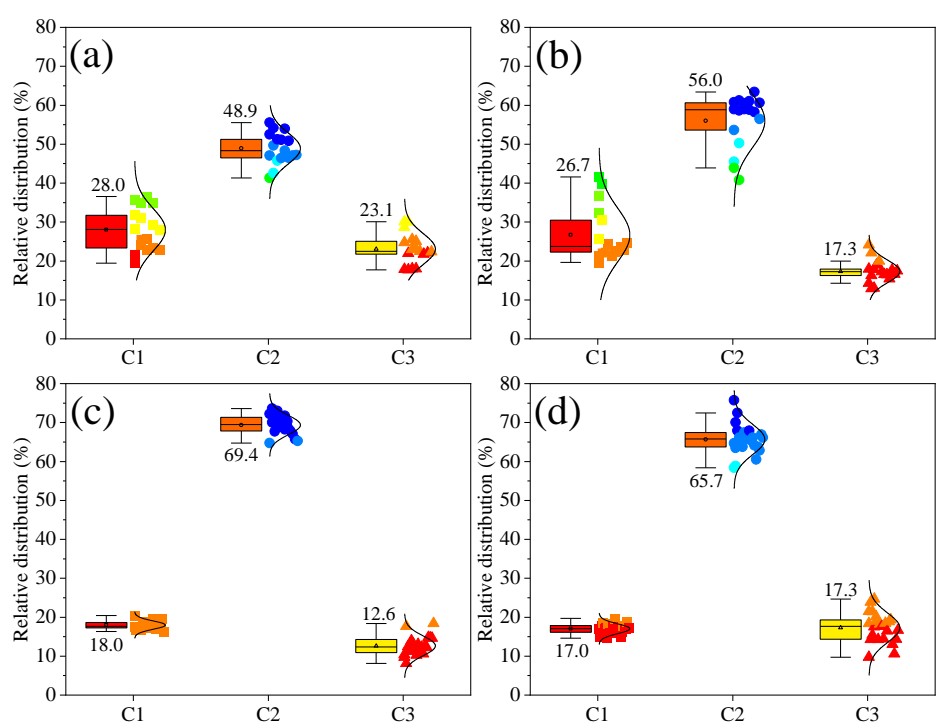

**Figure 4.** Relative contribution of individual fluorophores of atmospheric WSOM. (a:

Chuzhou (CZ); b: Guangzhou (GZ); c: wet season of GZ; d: dry season of GZ)