# Peer review of "Technical note: Identification of chemical composition and source of fluorescent"

_Atmospheric Chemistry and Physics, 2022_

## Editor Comment (EC1)

**Additional comments on acp-2022-676**

This is a well-prepared manuscript; there is also some discussion, but not much on atmospheric implications of authors' results. However, their findings provide new insights for the analysis of chemical properties and sources of atmospheric flourophores using the excitation-emission matrix fluorescence spectroscopy.

Besides the comments of the two experts in the field, I have few additional suggestions (lines in MS-version2):

Line 49: Some newer references on BrC organic compounds could be added (e.g., Frka et al., Chemosphere 2022, 299(11), 134381).

Line 67: For flourescent components in aerosol particles, reference of Ma et al. (Chemosphere, 2022) should be added.

Line 127 & lines 210-213: some references on typical atmospheric BrC compounds could be added (see e.g., Frka et al, 2022, Chemosphere).

Line 154: Error. Unit for the resistivity is MΩ·cm, so for Milli-Q water (ultra-pure) is 18.2 MΩ·cm.

Page 14 (end of paragraph): Please, check the references on formation of new secondary organic compounds (second generation) absorbing light at longer wavelengths (for example: Vidović et al., Environ. Sci. Technol. 2019, 53, 11195-11203; Vidović et al, Atmosphere 2020, 11, 131).

---

## Author Comment (AC1)

**Response to acp-2022-676-RC1**

Comment on acp-2022-676

Anonymous Referee #1

This manuscript investigated the EEM spectra of different types of strong light-absorbing organic compounds and water-soluble organic matter in different aerosol samples. The motivation is to broaden the application of the EEM-PARAFAC method to study atmospheric BrC. This manuscript is thoughtful and well-written. Below are some issues and comments for the authors to consider.

Re: Thanks for your valuable criticisms and comments, which are of great help for improving the quality of the manuscript. Based on your comments, we have revised the manuscript and provided a point-by-point response to all the comments. The detailed replies are as follows.

Introduction: some more description of WSOC should be involved in the introduction.

Re: Thanks. This is a good comment. In the present manuscript, we have added some descriptions of atmospheric WSOC in the introduction. Please refer to Lines 49-54.

Line 83: In the water environment, we usually call WSOC as DOM.

Re: Thanks. We have corrected it to "DOM" in the present manuscript. Please refer to Line 87.

---

## Author Comment (AC2)

**Response to acp-2022-676-RC2**

Comment on acp-2022-676

Anonymous Referee #1

The excitation-emission matrix (EEM) fluorescence spectroscopy is a highly sensitive analytical technique for the identification of the chemical characteristics and sources of atmospheric chromophores. However, some explanation may be inaccurate because the identification of fluorescent components is mainly based on aquatic DOM. This study investigated the EEM spectra of different types of strong light-absorbing organic compounds and water-soluble organic matter in different aerosol samples, soil dust, and purified fulvic and humic acids and some novelty findings were obtained. For example, aromatic compounds containing nitro groups, which show strong absorption and are the major component of atmospheric brown carbon, exhibited no significant fluorescence. In addition, the fluorescent component 1 (235, 270/330 nm) is generally considered as protein-like groups, however the results of this study suggested that it is mainly composed of aromatic acids, phenolic compounds, and their derivatives, with only traces of amino acids in ambient WSOM. In all, this manuscript is well-written and the major data and their interpretation are scientifically sound. I have some suggestions for changes that should be made before the manuscript is published, to wit:

Re: We appreciated the reviewer for the valuable comments and suggestions, which are of great help for improving the quality of the manuscript. We have revised the manuscript based on the comments and provided a point-by-point response to all the comments and explained how the comments and suggestions by the reviewer were addressed in the current version of the manuscript.

The comments and suggestions that need to be addressed: (L=Line)

Abstract: the abstract should include quantitative information on the point that are

made.

Re: Thanks. We have added some quantitative information in the abstract. Please refer to Lines 40-45.

Line 28. "by .. " should be changed to "supplemented by an parallel factor (PARAFAC) modeling".

Re: Thanks. We have changed that to "supplemented by a parallel factor (PARAFAC) modeling". Please refer to Lines 28-29.

Line 35. The "fluorescent component 1" is confused. I suggest to add "PARAFAC derived" to define it.

Re: Thanks. We have added "PARAFAC derived" to define it in the present manuscript. Please refer to Line 37.

Line 41: The "fluorescent component 3" refer to C3?

Re: Yes. The "fluorescent component 3" refer to C3 in this study. Based on the comments of reviewer #2, this sentence has been removed in the present manuscript.

Line 98. What are "various properties"? The reference here is not appropriate, because it only focused on fluorescence properties of aerosol chromophores. I suggest the authors provide more relevant references.

Re: Thanks. The "various properties" here specifically refer to its light absorption properties. We are sorry for this inappropriate reference. Base on your suggests, we have added more relevant references in the present manuscript. Please refer to Lines 101-102.

References:

Lin, H., and Guo, L.: Variations in Colloidal DOM Composition with Molecular Weight within Individual Water Samples as Characterized by Flow Field-Flow Fractionation and EEM-PARAFAC Analysis, Environ. Sci. Tech., 54, 1657-1667, http://doi.org/10.1021/acs.est.9b07123, 2020.

Huang, R.-J., Yang, L., Shen, J., Yuan, W., Gong, Y., Ni, H., Duan, J., Yan, J., Huang, H., You, Q., and Li, Y. J.: Chromophoric Fingerprinting of Brown Carbon from Residential Biomass Burning, Environ. Sci. Tech. Let., 9, 102-111, http://doi.org/10.1021/acs.estlett.1c00837, 2021.

Jiang, H., Tang, J., Li, J., Zhao, S., Mo, Y., Tian, C., Zhang, X., Jiang, B., Liao, Y., Chen, Y., and Zhang, G.: Molecular Signatures and Sources of Fluorescent Components in Atmospheric Organic Matter in South China, Environ. Sci. Tech. Let., 9, 913-920, https://doi.org/10.1021/acs.estlett.2c00629, 2022.

Line 110. Please add "peaks" after "fluorescence".

Re: Thanks. We have added 'peaks' after 'fluorescence' in the present manuscript. Please refer to Line 115.

Section 2.3: Please provide how Raman and Rayleigh scattering was removed from EEM.

Re: Thanks. In this study, the Raman and Rayleigh scattering was removed following a method suggested by Stedmon and Bro (2008) and Murphy et al. (2013), i.e., the Raman and Rayleigh scattering was removed by subtracting Milli-Q water spectra and then insert zero value to the Raman and Rayleigh scattering region. We have added this information in text S4 in the supporting information.

References:

Murphy, K. R., Stedmon, C. A., Graeber, D., and Bro, R.: Fluorescence spectroscopy and multi-way techniques. PARAFAC, Anal. Meth., 5, 6557, https://doi.org/10.1039/c3ay41160e, 2013.

Stedmon, C. A., and Bro, R.: Characterizing dissolved organic matter fluorescence with parallel factor analysis: a tutorial, Limn. Ocean. Meth., 6, 572-579, https://doi.org/10.4319/lom.2008.6.572, 2008.

EEM. Line 155. Please give a detailed definition of "IFEs".

Re: In this study, IFEs is the acronym of "inner filter effect". We have added a detailed definition in the present manuscript. Please refer to Line 165.

Line 163-165. I want to know if all the samples were investigated with the PARAFAC modeling?

Re: In this study, only 77 atmospheric WSOM samples were investigated with the PARAFAC modeling. We have clarified that in the present manuscript. Please refer to Line 173.

Line 190-194. I suggest the authors provide some references to support this point.

Re: Thanks. We have added some references (Chen et al., 2002; Andrade-Eiroa et al., 2013) to support this point in the present manuscript. Please refer to Line 203.

References:

Andrade-Eiroa, Á., Canle, M., and Cerdá, V.: Environmental Applications of Excitation-Emission Spectrofluorimetry: An In-Depth Review I, Appl. Spec. Rev., 48, 1-49, https://doi.org/10.1080/05704928.2012.692104, 2013.

Chen, J., Gu, B. H., LeBoeuf, E. J., Pan, H. J., and Dai, S.: Spectroscopic characterization of the structural and functional properties of natural organic

matter fractions, Chemosphere, 48, 59-68, https://doi.org/10.1016/s0045-6535(02)00041-3, 2002.

Line 220: Please remove the "and aquatic DOM" from sentence.

Re: Thanks. We have removed that from this sentence. Please refer to Line 230.

Line 277. Please add some references to support the similarities to EEM of BB and CC WSOM.

Re: Thanks. We have added some references to support that in the present manuscript. Please refer to Line 288.

References:

Chen, Q., Li, J., Hua, X., Jiang, X., Mu, Z., Wang, M., Wang, J., Shan, M., Yang, X., Fan, X., Song, J., Wang, Y., Guan, D., and Du, L.: Identification of species and sources of atmospheric chromophores by fluorescence excitation-emission matrix with parallel factor analysis, Sci. Total Environ., 718, 137322, http://doi.org/10.1016/j.scitotenv.2020.137322, 2020.

Fan, X., Cao, T., Yu, X., Wang, Y., Xiao, X., Li, F., Xie, Y., Ji, W., Song, J., Peng, P., amp, apos, and an: The evolutionary behavior of chromophoric brown carbon during ozone aging of fine particles from biomass burning, Atmos. Chem. Phys., 20, 4593-4605, http://doi.org/10.5194/acp-20-4593-2020, 2020.

Cao, T., Li, M. J., Zou, C. L., Fan, X. J., Song, J. Z., Jia, W. L., Yu, C. L., Yu, Z. Q., and Ping, P. A.: Chemical composition, optical properties, and oxidative potential of water- and methanol-soluble organic compounds emitted from the combustion of biomass materials and coal, Atmos. Chem. Phys., 21, 13187-13205, http://doi.org/10.5194/acp-21-13187-2021, 2021.

Yang, Y., Qin, Y., Qin, J., Zhou, X., Xv, P., Tan, J., and Xiao, K.: Facile Differentiation of Four Sources of Water-Soluble Organic Carbon in

Atmospheric Particulates Using Multiple Fluorescence Spectral Fingerprints, Environ. Sci. Tech. Let., 9, 359-365, http://doi.org/10.1021/acs.estlett.2c00128, 2022.

Line 310. To support the opinion in L311-312 "Although the fluorescence intensities varied with different sites and seasons, the EEM spectra of WSOM were very similar", I suggested the authors to add more examples for the EEM fluorescence spectra of aerosols from different sites and seasons.

Re: Thanks for your important suggestion. We added more examples to support the opinion in the present manuscript. Please refer to Lines 321-326.

"Similar fluorescence bands have been previously identified in the EEM fluorescence spectra of WSOM from $PM_{2.5}$ in the cold and warm seasons in Aveiro, Portugal (Matos et al., 2015), the High Arctic atmosphere (Fu et al., 2015), Godavari, Nepal (Wu et al., 2019), Lanzhou and Xi'an, northwestern China (Qin et al., 2018; Chen et al., 2020), Chongqing, southwestern China (Wang et al., 2020), and Harbin, northeastern China (Ma et al., 2022)."

Line 304-315: What is the purpose of the EEM spectrum being divided into five regions in Figure 2, and also have a discussion about them. Also in Line 311-312: "Although the fluorescence intensities varied with different sites and seasons, the EEM spectra of WSOM were very similar". It may be more appropriate to replace "EEM spectra" with "shape of the EEM spectra".

Re: Thanks for your careful review. The fluorescence region integral (FRI) is a method to indicate the relative intensity of fluorescence in different regions (e.g., the five regions mentioned here) (Chen et al., 2003). Some FRI parameters can be used to indicate the physical/chemical properties of dissolved organic matter, such as the hydrophobicity parameter, humification parameter and Stokes shift parameter (Yang

et al., 2022). In our draft, we want to use the FRI method to interpret the EEM spectra of atmospheric WSOM. However, the result is poor and no significant differences were observed, so was not shown in the paper. We are sorry for the careless checking on the lines in Figure 2. In the present manuscript, we have removed that from Figure 1 and 2.

In addition, we have replaced "EEM spectra" with "shapes of the EEM spectra" in the revised manuscript. Please refer to Line 328.

References:

Chen, W., Westerhoff, P., Leenheer, J. A., and Booksh, K.: Fluorescence excitation - Emission matrix regional integration to quantify spectra for dissolved organic matter, Environ. Sci. Tech., 37, 5701-5710, https://doi.org/10.1021/es034354c, 2003.

Yang, Y., Qin, Y., Qin, J., Zhou, X., Xv, P., Tan, J., and Xiao, K.: Facile Differentiation of Four Sources of Water-Soluble Organic Carbon in Atmospheric Particulates Using Multiple Fluorescence Spectral Fingerprints, Environ. Sci. Tech. Let., 9, 359-365, https://doi.org/10.1021/acs.estlett.2c00128, 2022.

Line 319-320: "To better explain the various fluorophores in different atmospheric WSOM samples, the EEM spectra were resolved with the EEM–PARAFAC tool". This sentence is repeated with the meaning expressed above, suggest deleting this sentence.

Re: Thanks. We have removed this sentence in the present manuscript.

Line 326-328: It is written: "In general, these fluorescent components have been interpreted based on the knowledge of fluorescence characteristics of aquatic DOM.". However, in Line 329-332, the reference you cited was atmospheric WSOC.

Re: We are sorry for these wrong citations. In the revised manuscript, we have changed that with the references related to the fluorescence spectra of aquatic DOM. Please refer to Lines 342-346.

References:

Coble, P. G.: Characterization of marine and terrestrial DOM in seawater using excitation emission matrix spectroscopy, Marin. Chem., 51, 325-346, http://doi.org/10.1016/0304-4203(95)00062-3, 1996.

Liu, L., Song, C., Yan, Z., and Li, F.: Characterizing the release of different composition of dissolved organic matter in soil under acid rain leaching using three-dimensional excitation-emission matrix spectroscopy, Chemosphere, 77, 15-21, http://doi.org/10.1016/j.chemosphere.2009.06.026, 2009.

Wünsch, U. J., Bro, R., Stedmon, C. A., Wenig, P., and Murphy, K. R.: Emerging patterns in the global distribution of dissolved organic matter fluorescence, Anal. Meth., 11, 888-893, http://doi.org/10.1039/c8ay02422g, 2019.

Zhang, Y. L., Gao, G., Shi, K., Niu, C., Zhou, Y. Q., Qin, B. Q., and Liu, X. H.: Absorption and fluorescence characteristics of rainwater CDOM and contribution to Lake Taihu, China, Atmos. Environ., 98, 483-491, http://doi.org/10.1016/j.atmosenv.2014.09.038, 2014.

Zhou, Y., Yao, X., Zhang, Y., Shi, K., Zhang, Y., Jeppesen, E., Gao, G., Zhu, G., and Qin, B.: Potential rainfall-intensity and pH-driven shifts in the apparent fluorescent composition of dissolved organic matter in rainwater, Environ. Pollut., 224, 638-648, http://doi.org/10.1016/j.envpol.2017.02.048, 2017.

Line 334. Please add the "terrestrial" after "aquatic".

Re: Added. Please refer to Line 348.

Line 351-354: the reference is missing.

Re: Thanks. We have added some references in the present manuscript. Please refer to Lines 367-368.

References:

Bianco, A., Passananti, M., Deguillaume, L., Mailhot, G., and Brigante, M.: Tryptophan and tryptophan-like substances in cloud water: Occurrence and photochemical fate, Atmos. Environ., 137, 53-61, http://doi.org/10.1016/j.atmosenv.2016.04.034, 2016.

Song, T., Wang, S., Zhang, Y., Song, J., Liu, F., Fu, P., Shiraiwa, M., Xie, Z., Yue, D., Zhong, L., Zheng, J., and Lai, S.: Proteins and Amino Acids in Fine Particulate Matter in Rural Guangzhou, Southern China: Seasonal Cycles, Sources, and Atmospheric Processes, Environ. Sci. Tech., 51, 6773-6781, http://doi.org/10.1021/acs.est.7b00987, 2017.

Vione, D., Albinet, A., Barsotti, F., Mekic, M., Jiang, B., Minero, C., Brigante, M., and Gligorovski, S.: Formation of substances with humic-like fluorescence properties, upon photoinduced oligomerization of typical phenolic compounds emitted by biomass burning, Atmos. Environ., 206, 197-207, http://doi.org/10.1016/j.atmosenv.2019.03.005, 2019.

Mahamuni, G., Rutherford, J., Davis, J., Molnar, E., Posner, J. D., Seto, E., Korshin, G., and Novosselov, I.: Excitation–Emission Matrix Spectroscopy for Analysis of Chemical Composition of Combustion Generated Particulate Matter, Environ. Sci. Tech., 54, 8198-8209, http://doi.org/10.1021/acs.est.0c01110, 2020.

Line 387-390. Similar questions above. Can these two samples be representative? I suggest the authors summarize more samples in detail.

Re: Thanks. According to your suggestion, we have added more examples in the present manuscript. Please refer to Lines 401-406.

"Similar fluorescent substances have also been identified in the study of atmospheric aerosol fluorescent chromophores, such as the highly oxygenated HULIS in Nagoya, Japan (Chen et al., 2016a), Lanzhou, China (Qin et al., 2018), Xi'an, China (Chen et al., 2020), a haze event in Harbin (Ma et al., 2022), and humic-like compounds with more aromatic and unsaturated bond in Godavari, Nepal (Wu et al., 2019) and Tianjin, China(Deng et al., 2022)."

Line 388: Please explain the "HULIS-1" used here. HULIS-1 of what?

Re: In this study, the 'HULIS-1' is the highly oxygenated HULIS in atmospheric WSOM (Chen et al., 2016). We have clarified that in the present manuscript. Please refer to Line 402.

Line 389: It may be more appropriate to replace "component 2" with corresponding compound.

Re: Thanks. We agreed with your suggestion and have used "humic-like compounds with more aromatic and unsaturated bonds" to replace "component 2" in the revised manuscript. Please refer to Line 405.

Line 485-487. The conclusion on contributions of fluorophores within CZ WSOM was wrong.

Re: We are sorry for this clerical error. In the present manuscript, we have corrected that in the conclusion. Please refer to Lines 485-489.

---

## Author Comment (AC3)

**Response to acp-2022-676-RC3**

Comment on acp-2022-676

Anonymous Referee #2

In this study, EEM data of different types of strongly light-absorbing organic compounds, water-soluble organic matter (WSOM), soil dust, and purified xanthic and humic acids from different aerosol samples (combustion source samples and ambient aerosols) were investigated in a comprehensive manner using the EEM-parallel factor method. This work can be recommended for publication in Atmospheric Chemical and Physics after the authors address some issues as follows.

Re: Thanks for your important suggestions. These criticism and suggestions will greatly improve the quality of this manuscript. And we have revised the manuscript based on the comments and suggestion and provided a point-by-point response to all the comments and explained how the comments and suggestions by the reviewer were addressed in the current version of the manuscript.

Lines 129-130: What is the purpose of setting up soil samples? Please explain it in detail.

Re: Thanks for your comments. In this study, the soil samples were also selected and tested because it is also an important source of atmospheric WSOM (Chen et al., 2020). On the one hand, soil and/or dustfall, have been reported to be an important source of atmospheric BrC, such as in Xi'an, Northern China (Chen et al, 2020; 2021) and a suburban site in Athens, Greece (Vasilatou et al., 2017). On the other hand, the terrestrial humic-like fluorescent components were commonly attributed to the contribution from soil in many studies (Liu et al., 2019; Ge et al., 2021). Therefore, soil samples were also selected to test the EEM method in this study. We have added a brief description in the revised manuscript. Please refer to Lines 138-140.

References:

Chen, Q., Li, J., Hua, X., Jiang, X., Mu, Z., Wang, M., Wang, J., Shan, M., Yang, X., Fan, X., Song, J., Wang, Y., Guan, D., and Du, L.: Identification of species and sources of atmospheric chromophores by fluorescence excitation-emission matrix with parallel factor analysis, Sci. Total Environ., 718, 137322, https://doi.org/10.1016/j.scitotenv.2020.137322, 2020.

Chen, Q., Hua, X., Wang, Y., Zhang, L., and Chang, T.: Semi-continuous measurement of chromophoric organic aerosols using the PILS-EEM-TOC system, Atmos. Environ., 244, 117941, https://doi.org/10.1016/j.atmosenv.2020.117941, 2021.

Ge, Z., Gao, L., Ma, N., Hu, E., and Li, M.: Variation in the content and fluorescent composition of dissolved organic matter in soil water during rainfall-induced wetting and extract of dried soil, Sci. Total Environ., 791, 148296, https://doi.org/10.1016/j.scitotenv.2021.148296, 2021.

Liu, C., Li, Z. W., Berhe, A. A., Xiao, H. B., Liu, L., Wang, D. Y., Peng, H., and Zeng, G. M.: Characterizing dissolved organic matter in eroded sediments from a loess hilly catchment using fluorescence EEM-PARAFAC and UV-Visible absorption: Insights from source identification and carbon cycling, Geoderma, 334, 37-48, https://doi.org/10.1016/j.geoderma.2018.07.029, 2019.

Vasilatou, V., Diapouli, E., Abatzoglou, D., Bakeas, E. B., Scoullos, M., and Eleftheriadis, K.: Characterization of PM2.5 chemical composition at the Demokritos suburban station, in Athens Greece. The influence of Saharan dust, Environ. Sci. Pollut. Res., 24, 11836-11846, https://doi.org/10.1007/s11356-017-8684-3, 2017.

Lines 134: Additional details whether the blank $PM_{2.5}$ sample was collected, please add this information.

Re: Thanks. In this study, field blank samples were also collected during each

sampling period. We have added this information in Section 2.1 in the revised manuscript. Please refer to Lines 146-147.

Lines 411-412: "The relatively higher C3 content in CZ could be attributed to the comparatively high contribution of soil dust in the suburban region". Please provide more evidence for the higher contribution of suburban soil dust to atmospheric $PM_{2.5}$.

Re: Thanks. In this study, the CZ sampling site is located at a typical suburban area (see S1.4 in SI file). In general, ambient aerosol in suburban region may have more contribution from soil dust, as shown in some previous studies (Vasilatou et a;l., 2017; Wu et al., 2019). Moreover, in this study, the relative contribution of $Ca^{2+}$ in CZ $PM_{2.5}$ (1.8±1.2%) are higher than that in GZ $PM_{2.5}$ (1.5% ±0.8). Therefore, we can concluded that the higher contribution of soil dust in the suburban region. We have added these information in the present manuscript. Please refer Line 432-433.

References:

Vasilatou, V., Diapouli, E., Abatzoglou, D., Bakeas, E. B., Scoullos, M., and Eleftheriadis, K.: Characterization of $PM_{2.5}$ chemical composition at the Demokritos suburban station, in Athens Greece. The influence of Saharan dust, Environ. Sci. .Pollut. Res., 24, 11836-11846, https://doi.org/10.1007/s11356-017-8684-3, 2017.

Wu, L., Luo, X.-S., Li, H., Cang, L., Yang, J., Yang, J., Zhao, Z., and Tang, M.: Seasonal Levels, Sources, and Health Risks of Heavy Metals in Atmospheric $PM_{2.5}$ from Four Functional Areas of Nanjing City, Eastern China, Atmosphere, https://doi.org/10, 419, 10.3390/atmos10070419, 2019.

Section 3.4. The paper also mentions that some brown carbon fractions have strong absorbance but not fluorescence characteristics, so is it reasonable to analyze the relationship between absorbance and fluorescence using Pearson correlation coefficient?

Re: Thanks. We agreed with your comments. Numerous studies have demonstrated that some brown carbon fractions, such as nitroaromatic compounds (NACs) are the major light-absorbing fraction in the atmospheric BrC, which accounting for more than 60 % of the total light absorption intensity at 300–500 nm (Huang et al., 2021; Lin et al., 2017), however, most of the NACs did not exhibit any fluorescence. It is obvious that the fluorescent index only represent a part of BrC rather than total BrC, therefore, the analysis of the relationship between absorbance and fluorescence using Pearson correlation coefficient should be unreasonable. Based on your suggestion, we have removed this section in the present manuscript.

References:

Huang, R.-J., Yang, L., Shen, J., Yuan, W., Gong, Y., Ni, H., Duan, J., Yan, J., Huang, H., You, Q., and Li, Y. J.: Chromophoric Fingerprinting of Brown Carbon from Residential Biomass Burning, Environ. Sci. Tech. Let., 9, 102-111, http://doi.org/10.1021/acs.estlett.1c00837, 2021.

Lin, P., Bluvshtein, N., Rudich, Y., Nizkorodov, S. A., Laskin, J., and Laskin, A.: Molecular Chemistry of Atmospheric Brown Carbon Inferred from a Nationwide Biomass Burning Event, Environ. Sci. Tech., 51, 11561-11570, http://doi.org/10.1021/acs.est.7b02276, 2017.

In Introduction part, Section 3.1 and 3.3, some new references associated with WOSM molecular and chemical functional group profiles should be added to support the finding of this study, such as:

1). Light absorption properties and molecular profiles of HULIS in $PM_{2.5}$ emitted from biomass burning in traditional "Heated Kang" in Northwest China. Sci. Total Environ. 2021, 776, 146014.;

2). Seasonal and diurnal variation of $PM_{2.5}$ HULIS over Xi'an in Northwest China: Optical properties, chemical functional group, and relationship with reactive oxygen

species (ROS). Atmos. Environ. 2022, 268, 118782.;

3). Optical properties, chemical functional group, and oxidative activity of different polarity levels of water-soluble organic matter in $PM_{2.5}$ from biomass and coal combustion in rural areas in Northwest China. Atmos. Environ., 2022, 283.

4) Optical properties, molecular characterizations, and oxidative potentials of different polarity levels of water-soluble organic matters in winter $PM_{2.5}$ in six China's megacities. Science of The Total Environment, 2022, 853: 158600

Re: Thanks, these new references have been added in the revised manuscript. Please refer to Lines 51-54, 60, 424-425.

Line 451: The conclusion of this part is not prominent enough, suggesting a more in depth analysis and better conclusions.

Re: Thanks. According to your critical comments on "the relationship between absorbance and fluorescence" as shown above. We have removed this section.

Figure 3 was lost the legend, please redraw this figure.

Re: Thanks, we redraw the Figure 3 with completed legends. Please refer to the new Figure 3.

Please add the necessary comments for Figure 4, what does each line represent?

Re: Thanks for your comments. We have added a legend in Figure 4, the colored box represents the data range of 25%-75%, the horizontal line within the box represents the median line (50%), the error bar represents the 1.5 times the standard deviation, the circle in the box represents the mean value of the data, the triangles in the bottom and top represent the minimum and maximum values of the data. And dot in the right of the box represents the overall data coupled with Gaussian distribution line. In

addition, we also added some comments on it in the revised manuscript. Please refer to the revised Figure 4 and Lines 421-431.

The language overall is acceptable, except for a few places which fail to meet the required level, advice on Grammar from a native writer of English would be helpful.

Re: Thanks. We have asked an English expert to edit English in our manuscript.

---

## Author Comment (AC4)

Dear Professor Irena Grgić,

Appended is our revised manuscript entitled "Technical note: Identification of chemical composition and source of fluorescent components in atmospheric water-soluble brown carbon by excitation-emission matrix with parallel factor analysis: Potential limitation and application". My coauthors and I have made all necessary revisions in the current version of the manuscript according to the comments and suggestions provided by the editor and the two reviewers. We feel that the revised manuscript has much improved quality and more convincing evidence than the prior version.

Thank you and the two reviewers for your comments which we found greatly improve the quality of manuscript. In the reply file, we have given a point-by-point response to all the comments and explained how the comments and suggestions by the editor and reviewer were addressed in the current version of the manuscript.

Please let me know if you have any question about our revised manuscript and thank you again for your assistance.

Looking forward to hearing from you.

Best regards,

Jianzhong Song
Guangzhou Institute of Geochemistry,
Chinese Academy of Sciences
511 KeHua Street, Guangzhou 510640, PR China
E-mail: songjzh@gig.ac.cn

**Response to Additional comments on acp-2022-676**

This is a well-prepared manuscript; there is also some discussion, but not much on atmospheric implications of authors' results. However, their findings provide new insights for the analysis of chemical properties and sources of atmospheric flourophores using the excitation-emission matrix fluorescence spectroscopy.

Besides the comments of the two experts in the field, I have few additional suggestions (lines in MS version2):

Line 49: Some newer references on BrC organic compounds could be added (e.g., Frka et al., Chemosphere 2022, 299(11), 134381).

Re: Thanks for your suggestion. We have added some newer references on BrC organic compounds in the present manuscript. Please refer to Lines 49-54.

References:

Frka, S., Šala, M., Brodnik, H., Štefane, B., Kroflič, A., and Grgić, I.: Seasonal variability of nitroaromatic compounds in ambient aerosols: Mass size distribution, possible sources and contribution to water-soluble brown carbon light absorption, Chemosphere, 299, 134381, https://doi.org/10.1016/j.chemosphere.2022.134381, 2022.

Huang, S., Luo, Y., Wang, X., Zhang, T., Lei, Y., Zeng, Y., Sun, J., Che, H., Xu, H., Cao, J., and Shen, Z.: Optical properties, chemical functional group, and oxidative activity of different polarity levels of water-soluble organic matter in $PM_{2.5}$ from biomass and coal combustion in rural areas in Northwest China, Atmos. Environ., 283, 119179, https://doi.org/10.1016/j.atmosenv.2022.119179, 2022.

Ma, L., Li, B., Yabo, S. D., Li, Z., and Qi, H.: Fluorescence fingerprinting characteristics of water-soluble organic carbon from size-resolved particles

during pollution event, Chemosphere, 307, 135748, https://doi.org/10.1016/j.chemosphere.2022.135748, 2022.

Zhang, T., Shen, Z., Huang, S., Lei, Y., Zeng, Y., Sun, J., Zhang, Q., Ho, S. S. H., Xu, H., and Cao, J.: Optical properties, molecular characterizations, and oxidative potentials of different polarity levels of water-soluble organic matters in winter $PM_{2.5}$ in six China's megacities, Sci. Total Environ., 853, 158600, https://doi.org/10.1016/j.scitotenv.2022.158600, 2022.

Line 67: For flourescent components in aerosol particles, reference of Ma et al. (Chemosphere, 2022) should be added.

Re: Thanks for your suggestion. This is an important reference for the fluorescent components in ambient particles. We have added it in the present manuscript. Please refer to Line 71.

References:

Ma, L., Li, B., Yabo, S. D., Li, Z., and Qi, H.: Fluorescence fingerprinting characteristics of water-soluble organic carbon from size-resolved particles during pollution event, Chemosphere, 307, 135748, https://doi.org/10.1016/j.chemosphere.2022.135748, 2022.

Line 127 & lines 210-213: some references on typical atmospheric BrC compounds could be added (see e.g., Frka et al, 2022, Chemosphere).

Re: Thanks. According to your comments, we have added some references on typical atmospheric BrC compounds in the present manuscript. Please refer to Lines 134-135 and 223.

References:

Frka, S., Šala, M., Brodnik, H., Štefane, B., Kroflič, A., and Grgić, I.: Seasonal

variability of nitroaromatic compounds in ambient aerosols: Mass size distribution, possible sources and contribution to water-soluble brown carbon light absorption, Chemosphere, 299, 134381, https://doi.org/10.1016/j.chemosphere.2022.134381, 2022.

Huang, R.-J., Yang, L., Shen, J., Yuan, W., Gong, Y., Ni, H., Duan, J., Yan, J., Huang, H., You, Q., and Li, Y. J.: Chromophoric Fingerprinting of Brown Carbon from Residential Biomass Burning, Environ. Sci. Tech. Let., 9, https://doi.org/102-111, 10.1021/acs.estlett.1c00837, 2021.

Wang, X., Gu, R., Wang, L., Xu, W., Zhang, Y., Chen, B., Li, W., Xue, L., Chen, J., and Wang, W.: Emissions of fine particulate nitrated phenols from the burning of five common types of biomass, Environ. Pollut., 230, 405-412, https://doi.org/10.1016/j.envpol.2017.06.072, 2017.

Line 154: Error. Unit for the resistivity is MΩ cm, so for Milli-Q water (ultra-pure) is 18.2 MΩ cm.

Re: We are sorry for this mistake. In the present manuscript, we have corrected that to 18.2 MΩ cm. In addition, we also double check the full text to avoid the similar errors. Please refer to Line 164.

Page 14 (end of paragraph): Please, check the references on formation of new secondary organic compounds (second generation) absorbing light at longer wavelengths (for example: Vidović et al., Environ. Sci. Technol. 2019, 53, 11195-11203; Vidović et al, Atmosphere 2020,11, 131).

Re: Thanks for your suggestion. We have checked the references citation on the formation of new secondary organic compounds absorbing light at longer wavelengths and added some new references in the present manuscript. Please refer to Lines 312-313.

References:

Powelson, M. H., Espelien, B. M., Hawkins, L. N., Galloway, M. M., and De Haan, D. O.: Brown carbon formation by aqueous-phase carbonyl compound reactions with amines and ammonium sulfate, Environ. Sci. Tech., 48, 985-993, https://doi.org/10.1021/es4038325, 2014.

Vidović, K., Kroflič, A., Jovanovič, P., Šala, M., and Grgić, I.: Electrochemistry as a Tool for Studies of Complex Reaction Mechanisms: The Case of the Atmospheric Aqueous-Phase Aging of Catechols, Environ. Sci. Tech., 53, 11195-11203, https://doi.org/10.1021/acs.est.9b02456, 2019.

Vidović, K., Kroflič, A., Šala, M., and Grgić, I.: Aqueous-Phase Brown Carbon Formation from Aromatic Precursors under Sunlight Conditions, Atmosphere, 11, 131, https://doi.org/10.3390/atmos11020131, 2020.

Vione, D., Albinet, A., Barsotti, F., Mekic, M., Jiang, B., Minero, C., Brigante, M., and Gligorovski, S.: Formation of substances with humic-like fluorescence properties, upon photoinduced oligomerization of typical phenolic compounds emitted by biomass burning, Atmos. Environ., 206, 197-207, https://doi.org/10.1016/j.atmosenv.2019.03.005, 2019.

Yu, L., Smith, J., Laskin, A., George, K. M., Anastasio, C., Laskin, J., Dillner, A. M., and Zhang, Q.: Molecular transformations of phenolic SOA during photochemical aging in the aqueous phase: competition among oligomerization, functionalization, and fragmentation, Atmos. Chem. Phys., 16, 4511-4527, https://doi.org/10.5194/acp-16-4511-2016, 2016.

---

## Author Response (AR2)

**Response to comments:**

Additional comments:

As suggested by Referee #2, please complete figure caption (Figure 4) not only with a legend, but add some more information, that it can be immediately clear what it represents.

Re: Thanks. According to comments, we have added more information in the caption of Figure 4.

"**Figure 4**. Relative contribution of individual fluorophores of atmospheric WSOM. (a: Chuzhou (CZ); b: Guangzhou (GZ); c: wet season of GZ; d: dry season of GZ; the colored box represents the data range of 25%-75%, the horizontal line within the box represents the median line (50%), the error bar represents the 1.5 times the standard deviation, the circle in the box represents the mean value of the data, the triangles in the bottom and top represent the minimum and maximum values of the data, the dot in the right of the box represents the overall data coupled with Gaussian distribution line.)".